# A Comparison of an Adaptive Self-Guarded Honeypot with Conventional Honeypots

**Sereysethy Touch * and Jean-Noël Colin** 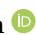

NaDI Research Institute, University of Namur, 5000 Namur, Belgium; jean-noel.colin@unamur.be
* Correspondence: sereysethy.touch@unamur.be

**Abstract:** To proactively defend computer systems against cyber-attacks, a honeypot system—purposely designed to be prone to attacks—is commonly used to detect attacks, discover new vulnerabilities, exploits or malware before they actually do real damage to real systems. Its usefulness lies in being able to operate without being identified as a trap by adversaries; otherwise, its values are significantly reduced. A honeypot is commonly classified by the degree of interactions that they provide to the attacker: low, medium and high-interaction honeypots. However, these systems have some shortcomings of their own. First, the low and medium-interaction honeypots can be easily detected due to their limited and simulated functions of a system. Second, the usage of real systems in high-interaction honeypots has a high risk of security being compromised due to its unlimited functions. To address these problems, we developed Asgard an adaptive self-guarded honeypot, which leverages reinforcement learning to learn and record attacker's tools and behaviour while protecting itself from being deeply compromised. In this paper, we compare Asgard and its variant Midgard with two conventional SSH honeypots: Cowrie and a real Linux system. The goal of the paper is (1) to demonstrate the effectiveness of the adaptive honeypot that can learn to compromise between collecting attack data and keeping the honeypot safe, and (2) the benefit of coupling of the environment state and the action in reinforcement learning to define the reward function to effectively learn its objectives. The experimental results show that Asgard could collect higher-quality attacker data compared to Cowrie while evading the detection and could also protect the system for as long as it can through blocking or substituting the malicious programs and some other commands, which is the major problem of the high-interaction honeypot.

**Keywords:** adaptive honeypot; self-guarded honeypot; reinforcement learning; *Q*-learning; conventional honeypot





## 1. Introduction

Cyber-attacks are probably one of the most important security threats that we are facing today. According to the Global Threat Intelligence Report 2021 [1], the attacks on finance, manufacturing and healthcare have increased significantly and they accounted for 62% in 2020 of the global attacks. There is a constant change in how an attack evolves in which we see new malware and trojans with new functions that try to exploit new and existing vulnerabilities, coin miners that exhaust the computing power and ransomware that target the bank sectors, among others. Understanding how those malware and trojans behave or finding the newly exploited vulnerabilities in a computer system or a loophole in the network is important, as they allow us to create a new signature for intrusion detection systems to detect them or fix these vulnerabilities.

To uncover these, a *honeypot* [2] is a decoy system that security researchers and professionals deploy on the network for the purpose of luring the attackers into attacking it to protect the real system and to collect intelligence data. However, these purposes can be achieved only if it can operate without being identified as a trap by adversaries; otherwise, its values are significantly reduced or completely diminished. Honeypots can

be classified by their degree of interaction that they provide to the attackers: *low*, *medium* and *high-interaction honeypots* [3–5].

Low-interaction honeypots (LiHP) simulate a limited function of a service or application just to receive the connection and provide very limited responses. On the other hand, medium-interaction honeypots (MiHP) are like LiHP, but they offer more functions to engage with attackers in a more convincing way. These low and medium-interaction honeypots are renowned for their simple implementation, easy deployment and maintenance and have a very low security risk because of their simulation nature. However, their downside is that they can be easily fingerprinted by attackers as a consequence of their limited and simulated functions. Contrarily to the LiHP and MiHP, high-interaction honeypots (HiHP) provide virtually unlimited functions by utilising a real operating system or a real application, which can capture higher-quality intelligence because the attacker can have access to all the functions of a real system. Despite these advantages, this type of honeypot is known to be difficult to deploy and maintain, and, most importantly, it carries a higher risk of the security being compromised.

To address the aforementioned shortcomings, we developed Asgard (The system was originally named Asguard, we changed it to Asgard to match the name of another system, Midgard, which we also developed) (Ref. [6]), which falls into a new class of honeypot called adaptive honeypots. These systems use machine learning techniques such as reinforcement learning (RL) [7] to learn through interaction to achieve their objective. RL is about an agent that learns to decide by taking action in its own environment, which is represented by its state. At each interaction, the agent receives a numerical value known as a reward from the environment that allows it to evaluate its own action. The objective of the agent is to find a strategy that can maximise its long-term accumulated rewards.

Asgard is an RL agent that uses actions such as allow, block and substitute to interact with its environment, which is represented by the Linux commands from attackers, to collect the attack data. However, instead of allowing the attackers to fully exploit the system after it is compromised, Asgard is built to block or substitute the execution of programs deemed malicious to limit the degree of being compromised. Asgard is designed to learn to balance between two objectives: (1) collect the attacker's tools and (2) protect itself from being deeply compromised.

In this paper, we compare Asgard and its variant Midgard [6] with two conventional SSH [8] honeypots: Cowrie [9] and a real Linux system. Midgard shares the same objectives as Asgard's; their difference lies in the way their reward functions are defined, which are used to train the RL agent during the learning process: Asgard uses the environment state, which is the Linux command and the action taken by the agent to define its reward function, while Midgard only uses the environment state in its reward function. Cowrie [9] is a MiHP that can emulate the SSH and Telnet protocols, and it is widely used to emulate Linux servers and IoT devices [10–12] to capture the attack on those systems. It allows attackers to log in and execute some Linux commands through a fake shell.

The goal of the paper is (1) to demonstrate the effectiveness of Asgard, which can collect the attacker's tools while evading detection, and keep the system safe for as long as it can through blocking or substituting the execution of malicious programs, and (2) the benefit of using the environment state and the action in RL to define the reward function to allow the agent to effectively learn its objectives. Our contributions in this manuscript are threefold:

- We describe the adaptive self-guarded honeypot Asgard that uses the environment state and action to define its reward function in the reinforcement learning problem that allows it to achieve its defined objectives.
- We describe its design and its implementation using a proxy-based architecture, which allows Asgard to use a real system as a honeypot.
- We compare the performance of Asgard from a real deployment with Midgard and two other conventional honeypots: Cowrie and a real Linux system.

The rest of the paper is organised as follows: background information and related work are presented in Sections 2 and 3, respectively. Section 4 describes the honeypot systems used in our experiments, and their design and implementation are described in Section 5. Section 6 presents the experimental results and lessons learned. Finally, a conclusion and future works are provided in Section 7.

## 2. Background

This section introduces the reinforcement learning approach, its formulation and learning methods. We also describe how an agent learns to trade off between exploration and exploitation. *Q*-learning is also presented; it is the algorithm that is used by our systems. We also present how the attack data from a honeypot can be used to build a state machine to derive the attacker's behaviour.

### 2.1. Reinforcement Learning

Reinforcement learning is a field of machine learning, but unlike the supervised and unsupervised learning techniques, it involves an agent that learns to make decisions by interacting with its own environment through the trial-and-error paradigm [7]. To allow the agent to learn, the environment needs to return a value, known as a reward, to the agent after it takes action. Based on this reward, the agent needs to find a strategy to maximise its accumulative rewards in the long term. The reward definition should represent the objective that we want the agent to learn. For example, if we want to teach the agent to learn how to park a car, the agent will be rewarded positively when it correctly parks the car, but it will be rewarded negatively when it fails. This learning process has been successfully used in various applications, such as the AlphaGo Zero system that beat a human in the game of Go after learning without human knowledge, just by playing against itself [13].

#### 2.1.1. Markov Decision Processes

The RL problem is modelled by Markov Decision Processes (MDP), as shown in Figure 1. At a time $t$, facing the environment, which is represented by a *state* $S_t$, the agent selects and executes an *action* $A_t$. In return, it receives a reward $R_{t+1} \in \mathbb{R}$ that allows it to evaluate its decision making. Subsequently, the agent now receives a new state $S_{t+1}$, and again, selects a new action $A_{t+1}$, then receives a new reward $R_{t+2}$, and continues, again and again, improving its behaviour [7]. The *Markov property* assumes that the *next action* only depends on the *current state* but not any other *previous state*.

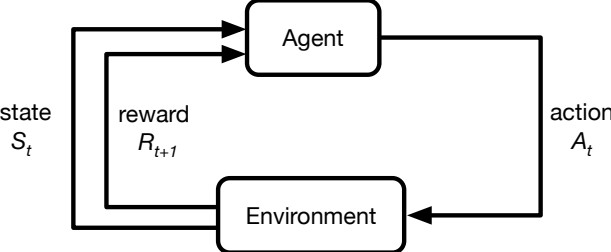

**Figure 1.** Reinforcement Problem–Interaction between agent and its environment.

Formally, an MDP is a tuple $< S, A, P, R >$ where

- $S$ is a (finite) set of states $s \in S$, which represents the environment,
- $A$ is a (finite) set of actions $a \in A$ that the agent can perform,
- $P_{ss'}^a$ is the probability of reaching state $s'$ from state $s$ after taking action $a$

$$P_{ss'}^a = P(S_{t+1} = s' | S_t = s, A_t = a) \tag{1}$$

- $R_s^a$ is a reward function that the agent receives when it is in state $s$ and is taking an action $a$.

Given this MDP, through the interactions between the agent and the environment, it produces a sequence of interactions called a *trajectory* which consists of a sequence of states, actions and rewards: $S_0, A_0, R_1, S_1, A_1, R_2, S_3, \ldots$ [7]. The goal of the agent is to find a policy $\pi$ to maximise its total reward, called a *return G*. Each received reward is not equally valued because the agent can value the immediate reward more compared to the delayed rewards that the agent receives far in the future. Thus, the return $G$ is the total discounted rewards from time-step $t$, which is defined as

$$G_t = R_{t+1} + \gamma R_{t+2} + \ldots = \sum_{k=0}^{\infty} \gamma^k R_{t+k+1} \tag{2}$$

where $\gamma \in [0, 1]$ is a discount factor.

The experiences can be derived from *episodic* tasks or *continuous* tasks. An episodic task has a start state and a terminating state, while a continuous one has no terminating state [7]. The agent uses these experiences to evaluate and improve its current policy.

In most of the problems of reinforcement learning, an agent learns to estimate the *state-value* $v(s)$ or a *state-action* value $q(s, a)$, also known as the *q*-value. These values provide an insight into the future rewards that the agent can receive. These rewards depend on how the agent decides to choose an action in a given state. The value of state and state-action following a policy $\pi$ are denoted, respectively, as $v_\pi(s)$ and $q_\pi(s, a)$. The policy that allows the agent to maximise their state value is called an *optimal policy*, denoted as $\pi^*$ [7].

We differentiate two groups of learning methods: *model-based* and *model-free* methods. The model-based method requires the model of the environment to be completely defined and available to search for an optimal policy, whereas the model-free method relies only on the experiences through interacting with its environment and collecting rewards to find the optimal policy. Among these algorithms, we also distinguish two types of learning policy: *on-policy* and *off-policy*. The former describes the group of algorithms that use their current learned policy to improve its own policy; whereas the latter exploits a different policy to improve its current policy [7].

### 2.1.2. Exploration and Exploitation Trade-Off

The early stage of the learning process of the agent is very important; its success relies heavily on acting randomly by trying some actions, also known as *exploration* and trying the best-learned actions so far, also known as *exploitation* to get more rewards. As a result, the agent has to balance between exploration and exploitation [7] because if the agent keeps exploring forever or always chooses the known best action, it will probably never find an optimal policy. One of the random policies that is widely used is $\epsilon$-greedy, in which we initially choose a very small probability *epsilon* that the agent randomly selects an action, while the agent acts *greedily* by selecting the best action based on its current learned policy for the probability of $1 - \epsilon$. During the learning process, the value of $\epsilon$ has to be gradually decayed such that the agent increasingly favours its learned policy by choosing the best action and eventually converges to the optimal policy.

### 2.1.3. *Q*-Learning

*Q*-learning is the learning algorithm that is used to estimate the *q*-value of the state-action pair to allow the agent to be given a state to decide on action. This algorithm was theorised by Watkins et al. [14]; it is an off-policy model-free algorithm that is based on a temporal difference (TD) [7]. The TD method updates the estimates of value functions as soon as there is new information available using an error signal called a TD *error* found between different time steps. In fact, *Q*-learning is a TD(0) because its update is made after one time step. The update of the *q*-value in a state $s$ while taking an action $a$, observing a reward $r$ and reaching a state $s'$ by acting greedily onward is given as follows

$$q(s, a) = q(s, a) + \alpha \left[ r + \gamma \max_{a'} q(s', a') - q(s, a) \right] \tag{3}$$

where $\alpha \in [0,1]$ is the learning rate, and $\gamma \in [0,1]$ is the discount factor. Watkins et al. [14] have proved that if the agent keeps visiting all state-actions indefinitely, it will converge to an optimal policy. To help the agent make a trade-off between exploration and exploitation of its actions, we use an $\epsilon$-greedy policy (cf. Section 2.1.2). Algorithm 1 gives a pseudo code of the *Q*-Learning.

---

**Algorithm 1:** *Q*-learning algorithm [7,14]

Initialise q(s, a) for all states *s* and actions *a*;
**foreach** *episode* **do**
    Initialise state *s*;
    **repeat**
        Choose *a* from *s* using $\epsilon$-greedy policy derived from q;
        Take action *a*, observe *r*, *s'*;
        $q(s,a) = q(s,a) + \alpha[r + \gamma \max_{a'} q(s',a') - q(s,a)]$;
        Replace *s* with *s'*;
    **until** *s is terminal*;

---

## 2.2. Attacker'S Behaviour

Ramsbrock et al. examined attack data from a Linux honeypot and built a state machine from the collected Linux commands to profile the attacker's behaviour [15]. To construct the state machine, they defined seven states as follows:

- **CheckSW**—'Check software configuration': this describes commands that the attacker uses to gather more information about the system's software and its users. The commands in this state are `w`, `id`, `whoami`, `last`, `ps`, `cat /etc/*`, `history`, `cat .bash_history`, `php -v`.
- **Install**—'Install a program': this describes the software installed or the process of installing a new software by an attacker. For example, an attacker can unarchive a downloaded file, followed by other filesystem related commands, such as copying, moving and deleting files. These commands can be `tar`, `unzip`, `mv`, `rm`, `cp`, `chmod`, `mkdir`.
- **Download**—'Download a file': this describes the commands used to download a remote file. These commands can be `wget`, `ftp`, `curl`, `lwp-download`.
- **Run**—'Run a rogue program': this describes when the attacker runs a program that was downloaded. These kinds of commands can be detected when the attacker precedes './' in front of the program name.
- **Password**—'Change the account password': this describes when the attacker wants to change the password of an account.
- **CheckHW**—'Check the hardware configuration': this describes commands that enable an attacker to gather more information about the system's hardware.
- **ChangeConf**—'Change the system configuration': this describes the operation of attackers that alters the system state by changing the environment variable, terminating running programs, modifying files, etc. The commands included in this state are `export`, `PATH=`, `kill`, `nano`, `pico`, `vi`, `vim`, `sshd`, `useradd`, `userdel`.

Rambsbrock et al. also suggested the state **no-op**, which corresponds to the commands that have no effects on the system, and they are `cd`, `ls`, `bash`, `exit`, `logout`, `cat`. The rest of the other commands are classified as **unmatched**, due to a typographical error.

## 3. Related Work

The history of honeypot usage can be traced back to the 1990s, where Cheswick described a story about a cracker known as Berferd that attacked his Internet gateway from a stolen account [16]. Cheswick had an idea of adding some fake services to a computer that had no production purpose rather just to confuse and jail his uninvited guest. Later, it was only in 2001 that L. Spitzner [2] properly defined 'a honeypot is a security resource whose value lies in being probed, attacked, or compromised'. Besides the classification

of honeypots by their level of interactivity as described in the Introduction (cf. Section 1), honeypots can also be classified by their purpose, their design or their distribution.

Cowrie [9] is an open-source project that is maintained by Michel Oosterhof, its code is based on the system called Kippo [17], and they both are written in Python. Cowrie can simulate SSH and Telnet servers and can be configured to simulate different versions of Linux distributions. Cowrie accepts an authentication mechanism via a password file, in which we can define a password for a username or a wildcard for any passwords and/or usernames. One of the remarkable features of Cowrie is that it offers an almost indistinguishable interactive bash shell from the real Linux bash shell that allows the attacker to execute some Linux commands. Once authenticated, the attacker can explore the Linux file system by listing its directory, viewing files, and performing other file system-related commands, such as `cp`, `mv`, `rm`, and if they want to change the password of the compromised account, they can run the command `passwd` interactively. Furthermore, the attacker can also view the hardware configuration by running the command `uname`, check its `uptime`, examine its network configuration via `ifconfig`, see how many other users are currently online, read all the running processes and download files from the Internet by using the commands `wget` or `curl`. However, Cowrie can be easily detected, Surnin et al. [18] showed in their detection models, despite invalid commands input, Cowrie always returns zero as an exit status. Vettel et al. [19] also demonstrated that it can be fingerprinted at the transport layer, while Morishita et al. used a signature-based detection to identify Cowrie [20].

Wagener et al. [21] was the first to introduce the adaptive honeypot that used reinforcement learning [7] to dynamically change its behaviour to engage with the attackers. They developed a honeypot called Heliza [22], which is a modified and vulnerable Linux server that can intercept system calls to control the execution of Linux commands. After the attacker broke into the system through the SSH service, they could request to execute a command, Heliza could decide whether the command should be *to* execute or *blocked* by returning a message of 'command not found'; it could also choose to *substitute* it by returning a predetermined result. Another action is *insult*, it is used to learn the identity of the intruder. To drive this decision-making process, Heliza used a reinforcement learning algorithm called SARSA, a model-free and on-policy method [7] similar to *Q*-learning, to find its strategy to achieve its objective. Heliza can be set to fulfil one of these two objectives: (1) to collect the attacker's tools, or (2) to waste the attacker's time. Each objective has its own reward function in which only the environment state, which is the Linux, commands was used. The authors showed that to achieve the first objective, the command `wget`, `sudo` and the newly downloaded programs should be allowed. For the second objective, the commands `wget` and `tar` should be substituted with an error message, and the command `sudo` should be blocked.

Inspired by Heliza, Pauna et al. developed RASSH [23] and QRASSH [24], but they replaced the Linux system with Kippo [17] and later with Cowrie [9], its successor. In addition to the four actions of Heliza, a new action, *delay*, was added to slow down command executions. To make decisions on commands, RASSH also used the same SARSA algorithm [7], while QRASSH used a Deep *Q*-Networks (DQN) [25], a variation of *Q*-learning by applying a deep-learning approach. RASSH, however, could only achieve the objective of collecting the attacker's tools by using the same reward function as Heliza's. For QRASSH, it had two different reward functions, namely, simple and complex. However, only the simple reward could be learned while the complex one required more time to train.

More recently, Dowling et al. used the same algorithm, SARSA, as Heliza and RASSH to build their honeypot system based on Cowrie to deal with automated malware. Its objective is to hide its identity from automated malware to increase the command transitions. In their system, they only used a three-action set: *allow*, *block* and *substitute*, and simplified its reward function by rewarding the agent for any command executions [26]. The authors showed that their system could collect four times more command transitions than a high-interaction honeypot.

Although these adaptive systems paved the way for the development of smart honeypots, they also suffer the same problems mentioned in the introduction. The system, like Heliza, can be considered a HiHP because it also uses the real Linux system as a honeypot as a result, it has a high risk of security. On the other hand, the systems that rely on Cowrie, which is just an emulator of a Linux server, can be easily detected [18–20] as previously described.

## 4. Description of Honeypot Systems

In this section, we will describe the honeypot systems that are used in our study to compare their performances: our adaptive self-guarded honeypots, namely Asgard and its variant Midgard [6]. These systems pose as a vulnerable Linux system that attackers can easily compromise to gain remote access to through the SSH protocol.

### 4.1. Definitions

For a simple and concise description in the next sections and throughout the rest of this paper, we will define the following terms.

- *An episode* is defined as an SSH connection that starts from the moment that the attacker connects to the honeypot until the moment that they disconnect from it. *An attack* or *an attack episode* refers to *an episode* that the attacker uses either (1) to execute commands or (2) to perform TCP/IP port forwarding. As we only focus on the former attack, as such, *an episode* or *an episode of attack* only refers to the attack of the command executions unless it is explicitly stated otherwise. It is also possible that we use the terms *episode* and *connection* interchangeably to avoid repeating the same words.
- *An attack sequence* refers to the sequence of commands that an attacker can request to execute during an attack. Different attacks can share the same attack sequence. An attack sequence is said to be unique if it belongs to a single attack, but if an attack sequence is shared by multiple attacks, for example, two attacks, we will refer to it as an attack sequence shared by 2 attacks. So on and so forth.

### 4.2. Asgard: Adaptive Self-Guarded Honeypot

Asgard is our adaptive honeypot system that we developed in [6], which distinguishes itself from the other adaptive honeypots because rather than allowing the attackers to freely attack and compromise the system, it learns to limit the risk of being compromised by blocking or substituting the execution of programs deemed malicious. The system is disguised as a vulnerable and fully functional Linux system for this reason, it is also considered as a high-interaction honeypot. The vulnerable aspect of this system is to allow easy remote access through the SSH protocol by using a brute force or a dictionary attack. After they gain access to the system, they could open an interactive shell session or simply request to execute Linux commands, such as in any normal Linux system. Whenever an attacker enters a command, the system intercepts and analyses it, and decides to either *allow*, *block* or *substitute* the execution of that command. To learn to decide on the action, Asgard is an RL agent that observes its environment state, which consists of the entered command only, and learns to take actions.

#### 4.2.1. Environment

The state of the environment is defined as the attacker's command input. In a typical attack, attackers will input a sequence of these commands, and each command can be mapped to one of the following sets of commands:

- $L$: a set of basic shell commands and some other installed programs during the initial system setup, $L = \{\texttt{cd}, \texttt{pwd}, \texttt{echo}, \texttt{cp}, \texttt{rm}, \ldots\}$,
- $D$: a set of download commands that can be used to download files or programs from external servers: $D = \{\texttt{wget}, \texttt{curl}, \ldots\}$,
- $C$: a set of custom commands that are not originally available in the system and have to be downloaded from external servers, all commands of this type will be mapped to an element *custom*, hence $C = \{\texttt{custom}\}$,

- $U$: a set of other inputs that cannot be mapped to any other above set; they can be an empty string, ENTER keystroke, ..., hence $U = \{\texttt{unknown}\}$.

  The final state of the environment is

  $$S = L \cup D \cup C \cup U \tag{4}$$

### 4.2.2. Actions

Asgard can take three actions: (1) *allow* is to execute the command, (2) *block* is to deny its execution and (3) *substitute* is to fake its execution. The reason is that *block* can make the attacker change their attack behaviour or have recourse to different commands when they are not available on the system, which will result in more command transitions. Another reason is that *block* can protect the honeypot from being compromised by preventing the execution of malicious commands. *Substitute* can also increase command transitions facing new or unknown programs. However, the action *insult* from Heliza [22] is not included because insulting an intruder seems like an obvious way to reveal its own identity. Furthermore, *delay* from RASSH [23] is not included either, since delaying the execution of a simple or known command can be used to fingerprint the system.

### 4.2.3. Reward Function

Asgard is designed to achieve two objectives: (1) to capture attacker's tools via download commands, and (2) to limit the risk of being compromised by preventing the execution of custom commands because the attackers generally resort to their downloaded custom tools to further compromise the honeypot and use it for their advantage. These objectives are rather opposing to the way a honeypot is normally intended. The reason is that we believe that once the honeypot is fully compromised and participates actively in the attack, its value is no longer significant. Asgard differentiates itself from the existing honeypots [22,23,26] because its reward function depends on (1) the environment state and (2) the action, while other systems only use the environment state in their reward functions. Asgard's reward given at a time-step $t$ is defined as the piecewise function below:

$$r_a(s_t, a_t) = \begin{cases} 1 & \text{if } s_t \in D \quad \text{and} \quad a_t \in \{allow\} \\ -1 & \text{if } s_t \in C \quad \text{and} \quad a_t \in \{allow\} \\ 0 & \text{otherwise} \end{cases} \tag{5}$$

The first objective can be achieved by rewarding the agent when it allows the attackers to download their programs. That is, when an attacker transitions to a download command $D$, and the action taken is *allow*, the agent is rewarded with 1. For the second objective, however, the agent gets punished instead when it chooses to allow the execution of the malicious programs. Consequently, when the attacker transitions to a custom command $C$ and the agent selects *allow* as the action, it receives $-1$.

### 4.3. Midgard: A Variant of Asgard

We developed a variant of Asgard to test a different reward function and thus orient the agent toward a different behaviour. Midgard, whose objectives are the same as Asgard's, but the main difference is that its reward function only depends on the environment state [6]; as such, its reward function is defined as follows:

$$r_m(s_t, a_t) = \begin{cases} 1 & \text{if } s_t \in D \\ -1 & \text{if } s_t \in C \\ 0 & \text{otherwise} \end{cases} \tag{6}$$

When the attacker transitions to a download command $D$, the agent is rewarded with 1 regardless of any actions taken by the agent, and when the attacker transitions to a custom command $C$, it is given $-1$ for any actions.

## 5. Design and Implementation

In this section, we discuss the design and implementation of our honeypot, based on a proxy-based architecture that addresses the shortcomings of MiHP and the HiHP.

### 5.1. Proxy-Based Architecture

Asgard and Midgard were developed using a proxy architecture [6], as can be seen in Figure 2. In this architecture, our honeypot system is the proxy system that poses as the SSH server to the attacker and also the SSH client to the real OpenSSH server [27]. As such, all the communication between the attacker and the real system can be captured, logged and modified by the proxy system.

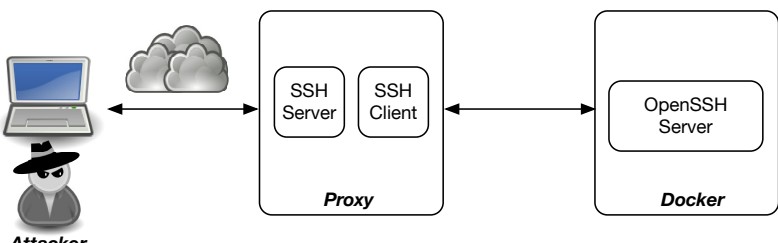

**Figure 2.** The architecture of proxy-based honeypots.

Even though the usage of a proxy is not new, as it has been used to detect web attacks or to defend web applications by using deceptive techniques [28–31], it has never been used in the SSH protocol to intercept and control the communication. The proxy-based architecture has several advantages: first, it allows a clear separation between our honeypot and the target system that we want to use as a honeypot. Secondly, the use of the real system behind the proxy allows us to avoid (1) its modification, which is necessary for the HiHP, and (2) the usage of an emulator like in MiHP that provides limited functions. Last but not least, using a container system such as Docker (https://www.docker.com/, accessed on 15 April 2022) to run the SSH server can contain the risk of malicious attacks.

### 5.2. Interaction between the Attacker and the Honeypot

Figure 3 shows the UML sequence diagram that illustrates the interaction between the attacker and Asgard as well as Midgard. The interaction starts when an attacker initiates a secure connection to the SSH proxy. After a connection is established, the attacker can try to log in by using a username and password. Once the authentication is successful, the attacker can send commands to the proxy. The proxy will examine the request and use it to query an action from the decision-making module, which implements the learning algorithm as described in Algorithm 1. If the action is *allow*, then the command will be forwarded for execution in the Linux container system. After the command is executed, its result is sent back to the proxy, which returns it back to the attacker. Alternatively, if the action is *block*, the proxy will return an error message 'command not found', and, finally, if the action is *substitute*, the command will not be executed, but the proxy will fake its execution by returning an empty message, which indicates that the command was successfully executed. After the command is executed, the execution result is sent back to the proxy to process and filter out some information before it is returned back to the attacker.

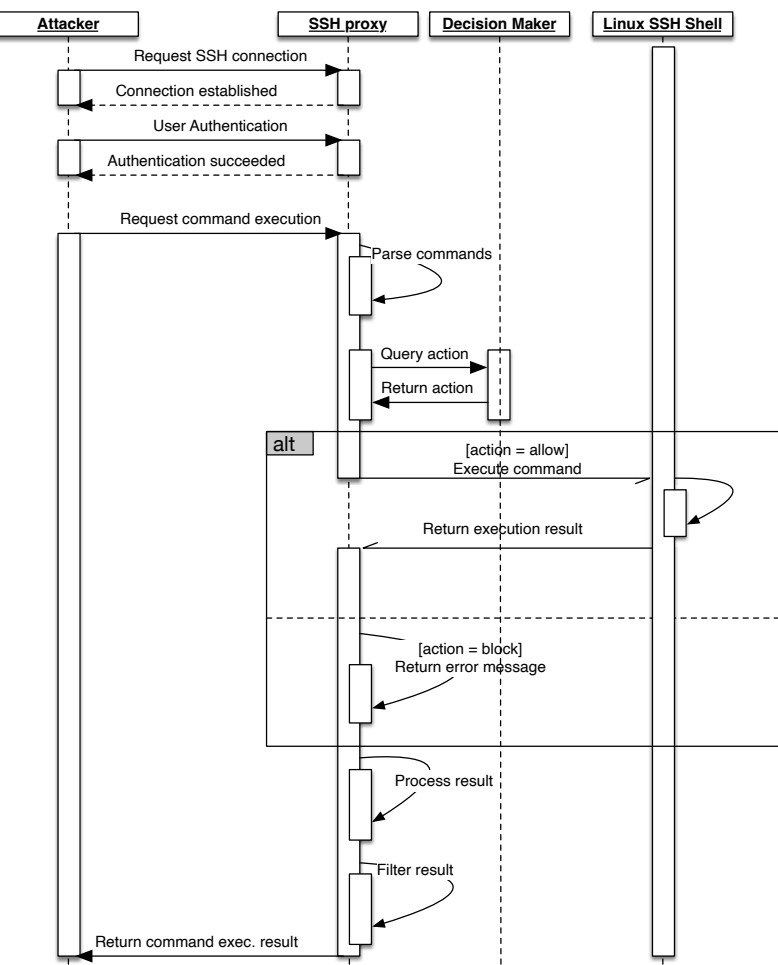

**Figure 3.** UML sequence diagram of the interaction between the attacker and Asgard/Midgard.

### 5.3. Linux Command Parsing

As shown in the sequence diagram Figure 3, after the request of the command execution, this command can be transmitted entirely to the target system and let the real shell handle it. In a normal execution flow, the shell will break it down into tokens of words and operators by obeying the quoting rules of the shell [32]. These tokens have to be parsed to construct a complete abstract syntax tree (AST) [33] that allows the shell to understand how to interpret them. The submitted command can be: a simple command optionally followed by arguments, compound commands (if, for, . . . ), a command with a redirection, a piped command, etc. For example, if it is a built-in shell command (e.g., `cd`, `break`, `exit`, etc.), it will be executed by the shell itself, otherwise, a new process is forked to execute an external command. If there is a redirection of the standard output of a command, the shell has to create a pipe and link it to the redirected file accordingly. For example, this command `ls -lh $(which ls)` was seen as a way to detect the Cowrie system (as the command substitution was not implemented in the old version of Cowrie), and its AST representation is shown in Figure 4. This AST is made up of a command node as a root node, which consists of three child-nodes: a word node `ls`, a word node `-lh` and a command substitution node. The child-node of the command substitution is also a command node comprised of a word node `which` and a word node `ls`. Our objective is to take action on each individual command that can be made up of the original command. As a result, the parsing has to be carried out by the proxy system to produce an AST, and from that, it derives a list of commands that can be analysed and decided one by one.

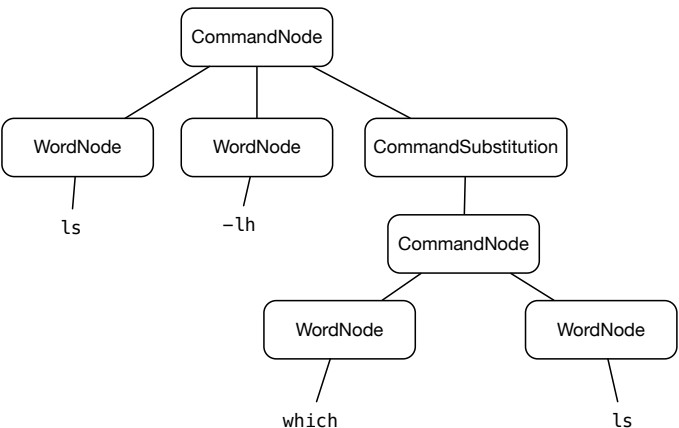

**Figure 4.** An AST represents a command 'ls -lh $(which ls)' that consists of a command substitution '$(which ls)'.

### 5.4. SSh Proxy

The SSH proxy module is developed by adding the proxy module and the decision-making module to the standard Cowrie honeypot [9]. Actually, this proxy can be implemented by any existing SSH library, such as Paramiko (https://www.paramiko.org/, accessed on 15 April 2022) or libssh (https://www.libssh.org/, accessed on 15 April 2022), but for convenience reasons, we used Cowrie as a base system as it already has a usable authentication service for an SSH server and also implements some useful functions, such as log and file downloads. The proxy module consists of an SSH server of Cowrie and an SSH client that is based on the Python Twisted Conch (https://twistedmatrix.com/, accessed on 15 April 2022) used by Cowrie. The decision-making module is implemented in Python3.

### 5.5. Decision-Making Module

The decision-making module provides two functions: (1) to update the agent's model given an episode and (2) to query the action given the state of the environment. The update function implements a slightly different version of Algorithm 2, as the model is updated online and for a single episode rather than a list of episodes in a batch mode. The second function is the action selection, which follows a $\epsilon$-greedy policy (cf. Section 2.1.2) derived from the current $q$-values, and its algorithm is as follows:

---

**Algorithm 2:** Action selection using the $\epsilon$-greedy policy to balance between exploration and exploitation

---

**Input**: $q$ function, $s$ an environment state representing the command, *actions* a list of actions, $\epsilon$;

$r$ = random(0,1);

**if** $r < \epsilon$ **then**

    action = random(*actions*);

    **return** *action*;

**else**

    action = $argmax_a q(s, a)$;

    **return** *action*;

---

### 5.6. Command Execution

To parse the shell command, we integrate *bashlex* [34], a Python library, which is a Python port of the parser used internally by GNU bash. From the AST that represents the input command string from the attacker, we produce a list of command objects that describe the type of the command, the command name, its arguments, etc. After that, for

each command in the command list we query the decision module about the action, which can be *allow*, *block* or *substitute*, and then execute the command based on the returned action. Algorithm 3 illustrates the process of executing a command.

---

**Algorithm 3:** Command execution

---

**Input**: a command string;
**Output**: result of the command execution;
*ast* = obtain from bashlex;
*command_list* = transform(ast);
**foreach** *command in command_list* **do**
    action = decision_module(command);
    result = execute_command(action, command);

---

## 6. Experimental Results

In this section, we will describe the performance of each system based on the attack data obtained from their real deployment and what we learned from this experiment.

To evaluate the performance of each system, we use the following criteria:

- The quantity and the quality of the attack data received: the number of attack episodes, the number of commands, which include the minimum, maximum and average number of commands executed per episode, and the number of attack sequences;
- The number and type of collected files;
- The attacker's behaviour;
- The $q$-values, which show how each adaptive honeypot learns its objectives.
- The number of incidents that each system was reported to participate in the attacks;
- The number of human attackers that each system lures into attacking it.

Before we present our experimental results, first, we will describe how the honeypot systems were set up and deployed and also how some hyperparameters were chosen to train both Asgard and Midgard systems for the experiment.

In addition to our honeypot systems Asgard and Midgard, we also set up a high-interaction honeypot, which we named Aster, using a plain, standard Linux server behind a proxy system similar to the Asgard's and Midgard's; however, in the Aster case, the proxy system is only used to authenticate users and record all communications. Aster is also vulnerable to the same brute force attack. We deployed each of these three systems, Asgard, Midgard and Aster, using two separate Docker containers: one for the proxy system and one for the OpenSSH server, on three different virtual machines running Linux Debian 10. The OpenSSH server was built from the official Linux Debian 10 docker image. The fourth platform, Cowrie, was also deployed as a Docker container on another virtual machine. We modified its default configuration, such as hostname, OpenSSH version and kernel information, to match that of the host system. To train Asgard and Midgard, these are the values of the hyperparameters that were used: $\epsilon = 0.5$ and its decay rate of 0.99984 for each update of the model—which amounts to 10K updates—until it reaches the minimum value of 0.1, $\alpha = 0.01$ and $\gamma = 0.99$. The systems were deployed for around 100 days from early 11/2021 until early 03/2022 on the public university network. The systems were stopped and restarted sporadically.

At the initial deployment, the systems were allowed to run at a maximum speed allocated for the virtual machines, but after noticing that our honeypots were frequently compromised and exploited to mine cryptocurrencies, we limited their CPU consumption as well as the network bandwidth. In each attack episode, we recorded the information related to the attack, such as network connection, session number, key exchange algorithms, SSH client version, connection time, username, password, entered commands, TCP/IP Port Forwarding requests [8] and payloads, connection duration and downloaded files. The update of the agent's model is conducted online; that is, its model is being updated and queried at the same time. The update of the model happens after each episode. However, it

is also possible that during the same episode, the model is updated multiple times because in the same SSH connection, there can be multiple channels open to request the command executions, which can also be considered as an episode.

### 6.1. Collected Attack Data

Table 1 shows the general statistics of the attack data that our honeypots received. Cowrie received the most attack episodes, while Aster received the least. Among those attack episodes, for Asgard, Midgard and Aster, only around 8% are really used for command execution, while the proportion reaches almost 17% for Cowrie; the rest are TCP/IP port forwarding requests. Even though Cowrie attracted the most attacks and also had the highest total number of commands, by closely looking at the average number of commands that each honeypot received per episode, Cowrie performed poorly, as it received only 5.51 commands on average compared to 18.48 commands for Aster, 9.68 for Asgard and 8.2 for Midgard. By examining the attack data, Cowrie received a large amount of the same single command echo -e "\x6F\x6B", which accounted for almost 51% of the total of attacks, which can explain its low average number of commands. These results confirm that the usage of a real system still performs best in terms of the average number of commands run per episode, although the adaptive honeypots were not far behind.

**Table 1.** Description of attack data.

|  | Asgard | Midgard | Cowrie | Aster |
|---|---|---|---|---|
| Number of Episodes | 395,692 | 417,606 | 452,047 | 109,134 |
| Number of Command Execution Episodes | 30,287 (7.65%) | 29,668 (7.10%) | 74,963 (16.58%) | 8310 (7.61%) |
| Number of TCP/IP Port Forwarding | 365,405 (92.34%) | 387,938 (92.89%) | 377,084 (84.41%) | 100,824 (92.38%) |
| Number of commands | 293,847 | 243,711 | 413,180 | 152,633 |
| Minimal number of commands | 1 | 1 | 1 | 1 |
| Maximal number of commands | 289 | 52 | 56 | 79 |
| Average number of commands | 9.70 | 8.21 | 5.51 | 18.36 |

Since most of the attacks were from automated programs and were very targeted, they mostly shared the same attack sequence. Let us be reminded that an attack sequence refers to a sequence of commands that are executed in an attack (cf. Section 4.1). For example, the three attack episodes below were captured in our honeypot and targeted Hive OS (https://hiveon.com/, accessed on 15 April 2022), which is software to manage a mining farm. In attacks 1 and 2, the attackers first wanted to change the password of the user managing this farm by using the command hive-passwd followed by the new password. Next, they wanted to terminate any active user sessions by killing the processes Xorg and x11vnc. These two attack sequences can be considered different or same. There are two methods that we can take into account: first, if we consider the commands with their arguments, then they are different, and, second, if we only consider the commands without their arguments, then they are the same because the two sequences of commands are the three commands sudo. However, attack 3 is different because of the command uname at the end.

```
# Attack 1
sudo hive-passwd specia4lbro123456; sudo pkill Xorg; sudo pkill x11vnc
# Attack 2
sudo hive-passwd tsrouble123; sudo pkill Xorg; sudo pkill x11vnc
```

```
# Attack 3
sudo hive-passwd j65kj675; sudo pkill Xorg; sudo pkill x11vnc; uname -a
```

Hence, we decided to apply the second method to find all the sequences of commands that were shared among the attack episodes, and we divided them into four groups of the command sequence based on the number of times that they were shared, this can give a good indication of the performance of each honeypot. According to Table 2, the number of the attack sequences found in the attack data is relatively small compared to that of the attack episodes: around 200 for Asgard and Midgard, 153 for Cowrie and 149 for Aster. However, if we look at the number of the unique attack sequences, they clearly show that the adaptive and the high-interaction honeypots outperformed the medium-interaction honeypot Cowrie, and Aster received the greatest percentage of unique attack sequences. Looking further at the attack sequences that were shared more than three times, Cowrie had the highest percentage of the redundant commands.

With these results, we could show that our adaptive honeypot Asgard always performs better than the medium-interaction honeypot but not better than the high-interaction honeypot.

**Table 2.** An attack sequence is the sequence of commands that were executed in an attack. It is unique when it belongs to a single attack. If it is shared by two attacks, it is referred to as an attack sequence shared by two attacks, etc.; this table shows the percentage and the actual number of attack sequences that each system received.

| | Asgard | Midgard | Cowrie | Aster |
|---|---|---|---|---|
| Unique attack sequences | 30.37% (65) | 24.62% (49) | 22.88% (35) | 36.91% (55) |
| Attack sequences shared by 2 attacks | 7.01% (15) | 10.55% (21) | 12.42% (19) | 11.41% (17) |
| Attack sequences shared by 3 attacks | 7.01% (15) | 5.03% (10) | 3.92% (6) | 4.03% (6) |
| Attack sequences shared by more than 3 attacks | 55.61% (119) | 59.80% (119) | 60.78% (93) | 47.65% (71) |
| **Total of attack sequences** | **100% (214)** | **100% (199)** | **100% (153)** | **100% (149)** |

### 6.2. Collected Files

After these systems were compromised, the attackers downloaded files or programs from various sources so that they could use them to further exploit the system. The downloaded file types are *plain text*, *executable*, *shell script*, *Perl script*, *Python program*, *HTML*, *archived* (tar) and *JSON*. After removing *plain text*, *HTML* and *JSON* files, the last row of Table 3 indicates the total number that each honeypot received, and they are not substantially different. However, Aster seemed to collect the lowest amount of files, which could be explained by the lowest number of connections it received (cf. Table 1).

To understand their content, we submitted the SHA-256 hashes of these files to www. virustotal.com (accessed on 15 April 2022), which analyses and keeps records of files from over 70 antivirus scanners to understand their contents. If virustotal.com has the record of the hash of a submitted file, it will return its information, such as *size, suggested threat, reputation, first submission date, last submission date* and *labels from antivirus scanners*. We aggregated these results by using their common generic type, for example, *malware*, to describe their specific types, such as *trojan.linux/mirai* or *trojan.linux/xorddos*.

Table 3 shows the type breakdown of all the downloaded files across all systems, and we found that the attackers generally infected the honeypots with malware, mostly of Mirai variants. They also exploited the honeypots to mine cryptocurrencies by predominantly using the XMRig (https://xmrig.com/, accessed on 15 April 2022), a cross-platform miner. Apart from Aster, Asgard, Midgard and Cowrie were installed with botnets that could communicate with their C&C servers using Internet Relay Chat (irc). The attackers also used a *speedtest* program written in Python to gather the information on the network connection on the three high-interactions, and they also used a hack tool that includes a port scanner and some other tools.

Other attacks identified as *shell downloaders* are the attacks in which they execute commands such as `wget` or `curl` to download files before executing them. The last groups of attacks are shell scripts that generally run tests to look for some specific platforms and download different versions of the same program for different platforms before executing them.

**Table 3.** Different types of files that were downloaded by each honeypot.

|  | **Asgard** | **Midgard** | **Cowrie** | **Aster** |
|---|---|---|---|---|
| Malware | 19 | 23 | 26 | 15 |
| Miner | 5 | 5 | 3 | 3 |
| IRC Botnet | 4 | 4 | 3 | 0 |
| Speed test | 1 | 1 | 0 | 1 |
| Log cleaner | 1 | 0 | 0 | 0 |
| Hack tools | 1 | 1 | 0 | 1 |
| Shell downloader | 11 | 13 | 16 | 8 |
| Shellscript | 3 | 5 | 4 | 3 |
| **Total** | **45** | **52** | **52** | **31** |

### 6.3. Attacker'S Behaviour

As explained in Section 2.2, Ramsbrock et al. built states that represent each command to study the attacker's behaviour [15]. As the attack landscapes have evolved since then, we need to consider some new commands to take into account these changes. Below are the states followed by the newly added commands:

- **CheckSW**: `sshd -V`, `users`, `crontab`, `hostname`, `who`, `info`, `help`, `show`, `modelname`. Some are not the commands usually found on Linux and supposedly target a very specific operating system, such as FortiOS (FortiOS is the operating system managing the firewall FortiGate (https://docs.fortinet.com/, accessed on 15 April 2022). They are not predominantly found in the attack data, but we decided to add them for completeness.
- **Install**: `dos2unix`, `apt`, `apt-get`, `yum`, `scp`, `rpm`, `touch`, `make`. We included these commands as the attackers also install some software using Linux package managers and some other tools needed to install their programs.
- **Download**: no new commands added.
- **Run**: `python3`, `nohup`, `nc`, `nice`, `disown`. In addition to these commands, we also detect the usage of `sh` to execute a shellscript and a command pipeline `'|base -s arg'` that uses `bash` to execute a newly downloaded shellscript.
- **Password**: `chpasswd`, `hive-passwd`. The command `hive-passwd` is to change the user in the HiveOS system.
- **CheckHW**: `nvidia-smi`, `nvidia-info`, `amd-info`, `pci`, `free`, `lscpu`, `lspci`, `nproc`, `nvidia-smi`, `ip`, `dmesg`, `dmidecode`. Some are the commands that the attackers are

actively using to look for graphics processing units (GPU), such as Nvidia and AMD, which are widely used to mine cryptocurrencies.

- **ChangeConf**: `pkill, systemctl, killall, service, unset, set, usermod, halt, shellinabox`.
- **no-op**: `dir, echo, grep, egrep, cut, awk, uniq, screen, clear, head, sed, wc, tr, which, sleep`.

Table 4 summarises the number of commands that correspond to each state for each honeypot. The states 'no-op' account for the largest commands shares, which can be almost 50% for Cowrie and Aster; for Asgard and Midgard, the 'no-op' coverage are in line with the percentage (34.08%) as reported by Rambsbrock et al. in his work. The other commands are included in the states 'unmatched', which, in addition, to the misspelled commands, we also include newline, rarely used commands such as `bioset` and `busybox` and the commands `su` or `sudo` alone without arguments, together they made up less than 0.1%.

Overall, Asgard and Midgard shared the same percentage of the command states, while Cowrie and Aster also shared the same percentage of the command states. However, the adaptive honeypots managed to execute the download commands the most, accounting for around 10%, compared to only 3.49% for Cowrie and 0.97% for Aster.

**Table 4.** State machine coverage for each honeypot.

| State | Asgard | Midgard | Cowrie | Aster |
|---|---|---|---|---|
| CheckHW | 60,019 (20.43%) | 46,305 (19.00%) | 91,926 (22.25%) | 37,273 (24.42) |
| Install | 36,596 (12.45%) | 35,792 (14.69%) | 35,690 (8.64%) | 12,154 (7.96%) |
| Download | 26,310 (8.95%) | 28,431 (11.67%) | 14,406 (3.49%) | 1481 (0.97%) |
| Run | 20,123 (6.85%) | 20,606 (8.46%) | 12,079 (2.92%) | 1851 (1.21%) |
| CheckSW | 17,473 (5.95%) | 11,870 (4.87%) | 28,127 (6.81%) | 13,939 (9.13%) |
| ChangeConf | 16,876 (5.74%) | 20,599 (8.45%) | 15,091 (3.65%) | 4151 (2.72%) |
| Password | 9654 (3.29%) | 7003 (2.87%) | 16,851 (4.08%) | 5886 (3.86%) |
| (unmatched) | 143 (0.05%) | 224 (0.09%) | 43 (0.01%) | 89 (0.06%) |
| (no-op) | 106,653 (36.30%) | 72,881 (29.90%) | 198,967 (48.16%) | 75,809 (49.67%) |
| **Coverage** | **100%** | **100%** | **100%** | **100%** |

From the state definitions, we construct a state diagram that allows us to understand how the attacker behaved once they had compromised the system. Each edge is labelled with a transition probability from one state to another state. To calculate the transition probability, we divided the total number of outgoing transitions from that state by the number of outgoing transitions to other states. The thickness of the node is proportional to the number of commands that fit in that state. To make the diagrams easy to understand, not all transitions are shown; we only highlighted some important transitions to make the attacker's behaviour more apparent.

Figures 5–8 show the state diagrams of each honeypot system. Across all the systems, the most likely route that the attackers took was first to check system hardware information, followed by checking other system hardware information, before downloading some files or checking the software information, installing the downloaded files and/or downloading more files, before finally executing them. Most of the attacks were from automated scripts.

In 2007, Ramsbrock et al. found that the attackers started by looking for the software configuration, changing the password and gathering hardware information, before downloading a program, installing it and running it [15]. Here, the trend seemed to change, the attackers seemed to be looking for certain hardware suitable for mining cryptocurrency, rather than for certain software or operating systems to infect them with malware. For Cowrie, the attack episode usually ended right after checking hardware information; this could also explain why the average number of commands was low (cf. Table 1).



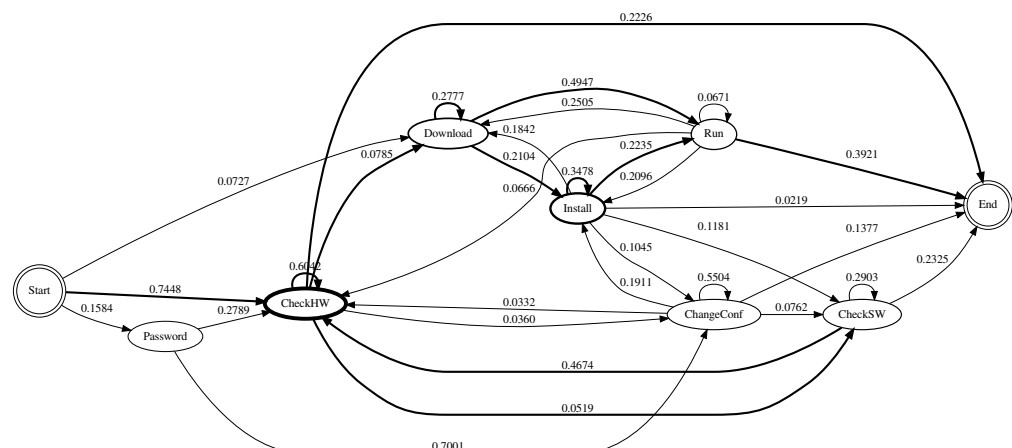

**Figure 5.** State diagram of the attacker's behaviour on Asgard.

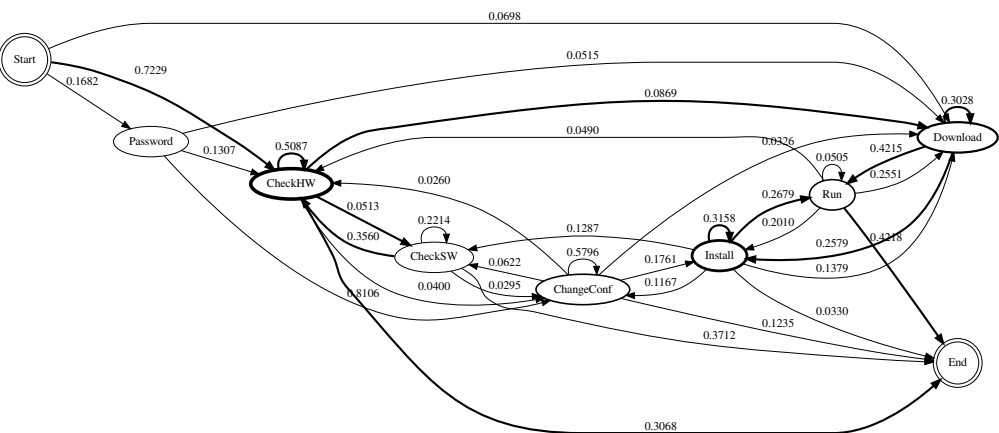

**Figure 6.** State diagram of the attacker's behaviour on Midgard.

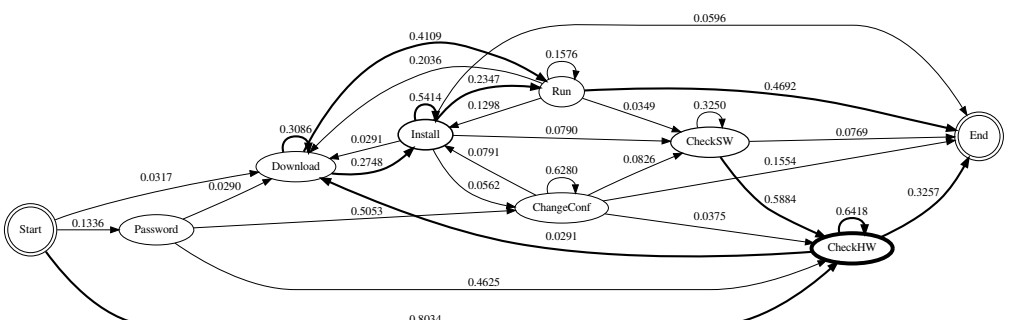

**Figure 7.** State diagram of the attacker's behaviour on Cowrie.

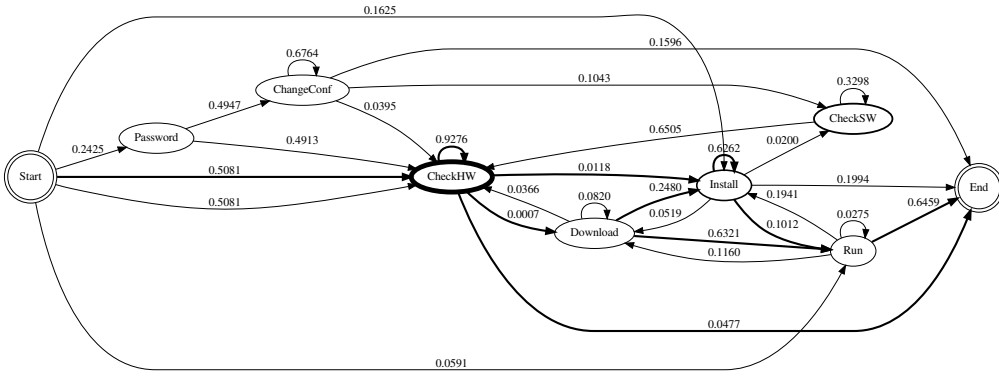

**Figure 8.** State diagram of the attacker's behaviour on Aster.

### 6.4. Q-Values

Tables 5 and 6 show the *q*-values of each command for Asgard and Midgard, respectively. The *q*-value indicates the preference of the action that the agent chooses given the command as an environment state. The selected action of each command is highlighted in bold. For Asgard, the action *allow* is selected for the download command `wget` with a *q*-value of 1.9696, which matches its first objective, while the selected action of the commands of `custom` is either *block* or *substitute* with a *q*-value of 0.0086, which also corresponds to the second objective of protecting the honeypot. These *q*-values nearly match the values that were reported in the results obtained from the attack simulation [6]. For Midgard, the *q*-value of the download command `wget` indicates that the action *substitute* should be selected, which is counterproductive. For the `custom` command, the action is to *allow*, which goes against the second objective of a self-guarded honeypot. These values enable us to confirm that by using the environment state and the action to define the reward function, the agent can successfully learn the intended objectives [6].

**Table 5.** *q*-Values of some commands of Asgard.

| Command | Allow | Block | Substitute |
| --- | --- | --- | --- |
| tar | **0.3927** | 0.1102 | 0.0983 |
| sudo | 0.0578 | **0.0765** | 0.0554 |
| chmod | **0.7317** | 0.2349 | 0.2655 |
| uname | 0.1026 | **0.1716** | 0.0943 |
| unknown | 0.0447 | **0.0454** | 0.0452 |
| custom | −0.4058 | 0.0086 | **0.0086** |
| ps | **0.1645** | 0.0703 | 0.0214 |
| wget | **1.9696** | 0.4153 | 0.3959 |
| bash | 0.0134 | 0.0135 | **0.0135** |
| sh | 0.1635 | 0.2392 | **1.1545** |

**Table 6.** *q*-Values of some commands of Midgard.

| Command | Allow | Block | Substitute |
| --- | --- | --- | --- |
| tar | **−0.2586** | −0.2589 | −0.2613 |
| sudo | −0.1699 | −0.1842 | **0.0957** |
| chmod | −0.4297 | −0.4491 | **0.3255** |
| uname | **0.9130** | 0.1210 | 0.1461 |
| unknown | −0.1308 | −0.0793 | **0.1480** |
| custom | **−0.9715** | −1.0743 | −1.0671 |
| ps | **0.2964** | 0.0391 | 0.0535 |
| wget | 1.0265 | **1.3700** | 1.0476 |
| bash | 0.0447 | **0.0449** | 0.0445 |
| sh | 0.1075 | 0.1350 | **0.9158** |

### 6.5. Reported Incidents

During the course of the deployment of these honeypots, a number of incidents were reported by the Belgian national Center for Cyber Security Belgium (CCB) that our honeypots were involved in various attacks. Table 7 reports the number of incidents originating from each system, and, obviously, Cowrie had no incident because it was just an emulator. For the three other systems, they used real Linux systems; as expected, they

could be used to attack other systems. Interestingly, Asgard received only one incident thanks to the way the system was designed to prevent the execution of custom programs. Once again, by blocking or substituting the custom programs, as shown by the *q*-values in Table 5, Asgard can prevent the system from being completely compromised.

**Table 7.** Number of incidents that were reported by the Centre for Cyber Security, Belgium.

|                      | **Asgard** | **Midgard** | **Cowrie** | **Aster** |
| -------------------- | ---------- | ----------- | ---------- | --------- |
| Number of incidents  | 1          | 16          | 0          | 6         |

*6.6. Human Attackers*

As we mentioned earlier, most of the attacks are from automated scripts, and it is quite rare to actually find real human attackers. Even so, our honeypots managed to attract some of them.

To spot them from the thousands of received connections, we analysed the logs and used some information derived from the SSH libraries used by the SSH client software. This could give a hint to *what* or *who* originated these attacks: a human attacker needs a Terminal or Pseudo Terminal (PTY) to type and visualise their commands, while automated programs do not. As a result, we chose some SSH client libraries that are usually used by human adversaries, such as *PuTTY* and *OpenSSH* for the Windows and Linux operating systems and combined them with some parameters of the terminal size, and together we narrowed down a number of connections that were originated from human attackers. More often, the same human attacker was seen connected multiple times to perform/resume their attack or to check users or running processes on the system; thus, these connections were only counted as from a single attacker.

Table 8 shows the number of human attackers on each system; once again, the adaptive and the high-interaction honeypots got more visits from the human attackers than the medium-interaction honeypot. By analysing those episodes, we found that most of the time, they only ran a few commands to check the running processes with the command `'top'` and also check the online users with the command `w`. The other command that is often used is `uptime` to check since when the system was running. We found one episode below on Midgard in which the attacker tried to download and install the mining programs.

**Table 8.** Number of human attackers that were actually connected to the honeypots.

|                            | **Asgard** | **Midgard** | **Cowrie** | **Aster** |
| -------------------------- | ---------- | ----------- | ---------- | --------- |
| Number of human attackers  | 10         | 14          | 1          | 10        |

```
# Midgard
lscpu
curl -s -L https://raw.githubusercontent.com/C3Pool/xmrig_setup/master/
setup_c3pool_miner.sh | bash -s 4AbDso7DmSj\dots
```

The presence of human attackers also emphasises the benefit of using the real system in the adaptive honeypots and in the high-interaction honeypot.

*6.7. Example of Interaction between Asgard and a Human Attacker*

The following attacks observed on Asgard demonstrated how it interacted with a human adversary after Asgard was compromised by defending itself through blocking and substituting some commands, which led to the attacker to finding another way to evade the protection mechanism.

After finding that their miner software was installed by their automated program, it *was not running* because apparently Asgard *deliberately blocked* its execution. As seen in the first episode below, it shows the commands and the actions taken by Asgard. In the

beginning, the attacker checked the system information, and the commands `uname` and `cat` were allowed. Next, they tried `python3`, which Asgard decided to substitute. Seeing no reply, they tried again, but this time with a typo `python33` and it was allowed, but the command did not exist. This time, they saw an error message, so they executed the correct `python3` only to learn that it was not available either. Then, the attacker proceeded to install `python3` and their intention was to get the host network information. One of the surprising twists in this attack is the *blocking* of the command `ls`, which forced them to find another way to list the content of the directory. During this time, they also checked the network information, but `ifconfig` was substituted by our system, they downloaded a *speedtest* program `v.py` from a malicious website [nasapaul.com](nasapaul.com) (accessed on 15 April 2022); due to the command `ls` was still blocked, they resorted to using the python program to execute `ls` in episode 2. In total, this attacker was connected 10 times before they could run their mining program XMrig; because Asgard continued to block the execution of `./xmrig` until there was an error in the implementation, the attacker managed to execute that program.

```
# Asgard: episode 1
uname -a                        -> allow
cat /etc/issue                  -> allow
python3                         -> substitute
python33                        -> allow, command not found
python3                         -> allow, command not found
sudo apt-get install python3    -> block, command not found
apt-get install python3         -> allow
python3 -> allow
...
>>> import socket                -> allow
>>> socket.getaddrinfo()
TypeError: getaddrinfo() missing 2 required positional arguments: 'host'
and 'port',
...
ls                              -> block
ifconfig                        -> substitute
ll                              -> allow, command not found
clear                           -> allow
ifconfig                        -> substitute
curl nasapaul.com/v.py          -> allow
ls                              -> block
ll                              -> allow, command not found
...

# Asgard: episode 2
python3 -> allow
import os -> allow
os.system('ls')                 -> parsing error, send "\n"
Ctrl+C
ls -> block
```

*6.8. Lessons Learned*

After running these experiments and analysing the collected attack data, we identify the following key takeaways:

- Although the high-interaction honeypot could potentially achieve the best result, it is at the cost of a high risk of being completely compromised (cf. Section 6.5). A system such as Asgard, however, did not allow the attacker to immediately compromise the system because it tried to keep the system safe for as long as it can by blocking and substituting malicious programs and some other commands that can lead to their execution (cf. Section 6.4).

- Another benefit of using the real system as a honeypot is that it can collect higher-quality attack data than the medium-interaction honeypot Cowrie. Despite Cowrie's largest attack numbers received, they contained a lot of short and redundant sequences of commands (cf. Section 6.2). More strong evidence that underscores the benefit of using a real system is its ability to attract human attackers (cf. Section 6.6) far more than the medium-interaction honeypot. This was also mentioned in Dowling's work, in which Cowrie did not get any human attackers [26].
- Even though Cowrie did not achieve a good result compared to the other systems, it also showed how a medium-interaction honeypot can still play an important role in capturing good attack data, as it was demonstrated by no substantial difference in the number of files collected by all the systems (cf. Section 6.2).
- There is a continuous trend that the attackers use the compromised machines to mine cryptocurrencies rather than infecting the system with malware, and the attacks are predominantly from automated programs.

## 7. Conclusions and Future Works

In this paper, we presented our design and implementation of a self-guarded adaptive honeypot for the SSH protocol and compared two variants of it with Cowrie, a widely used medium-interaction honeypot and a high-interaction honeypot, Aster, that we also set up for this comparative study.

The experimental results show that the adaptive self-guarded honeypot Asgard performs best by leveraging Reinforcement Learning to learn to compromise between collecting attack data and keeping the system safe. This shows that Asgard can be deployed with a lesser risk of security than the high-interaction honeypot. Another benefit that is worth repeating is the usage of the real system as a honeypot, thanks to the proxy-based architecture and implementation. This can avoid the problem of the medium-interaction honeypot being fingerprinted by the adversaries because of their limited and simulated functions.

The resulting $q$-values clearly show that the coupling of the environment state with the action in the reward function helps Asgard effectively learn its two objectives, as opposed to Midgard, which only uses the environment state to define its own reward function.

Nevertheless, there are still some limitations to our Asgard system:

- Its reward function encourages the agent to block or substitute the execution of malicious programs; this can prevent us from observing more complex attacks that could happen after the execution of these programs. The attacker can also use this feature to fingerprint our system by executing a test program.
- The implementation of our proxy system is still limited, as during the attacks, we were not able to deal with all of the command types, especially some complex commands such as pipeline commands and compound commands.
- The experiments were conducted only one time, and on the same network; as a result, we could not draw a definitive answer to all of our claims. It could have been tested in more different locations.
- Adaptive honeypots can also be subject to being fingerprinted by the adversaries by assuming all the honeypots are adaptive, in this case, they can launch 'blind attacks' by creating fake attacks to fingerprint these systems [35].

As for our future works, we intend to extend our adaptive self-guarded honeypot to enrich its description of the environment state by including the commands and their arguments, and also information about the honeypot system itself, such as CPU, memory and network flow. Indices of compromise coming from external IDS could also be used to create a richer state. As mentioned in the limitations, we see the potential by changing the static reward function to a more dynamic reward function to allow the execution of malicious programs for a certain degree of risk, which will bridge the gap between Aster and Asgard, and this, in turn, could achieve a better result. Another direction that we want to work on is to deploy several adaptive self-guarded honeypots in different locations and

allow them to exchange and exploit their learned models. We believe that this will speed up the learning process.

**Author Contributions:** Conceptualisation, S.T. and J.-N.C.; methodology, S.T.; software, S.T.; validation, S.T. and J.-N.C.; investigation, S.T.; data curation, S.T.; writing—original draft preparation, S.T.; writing—review and editing, J.-N.C.; visualisation, S.T.; supervision, J.-N.C. All authors have read and agreed to the published version of the manuscript.

**Funding:** This research received no external funding.

**Institutional Review Board Statement:** Not applicable.

**Informed Consent Statement:** Not applicable.

**Data Availability Statement:** The data presented in this study are available on request from the corresponding author. The data are not publicly available due to privacy concerns.

**Conflicts of Interest:** The authors declare no conflict of interest.

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
