# Peer review of "A Comparison of an Adaptive Self-Guarded Honeypot with Conventional Honeypots"

_applsci, doi:10.3390/app12105224_

Round 1

Reviewer 1 Report

The authors present a comparative study of four honeypot systems, of which two are adaptive. An implicit goal seems therefore to show whether adaptive ones perform better than the non-adaptive ones. The paper is in a sense relatively well written, but in some areas lacking when it comes to clarity and conciseness, and there are some other shortages.

Although four honeypot systems are considered, the abstract gives the impression that only one is considered (Asgard). The following sentence is incomprehensible, and must be rewritten and split: 'With an abundance of these systems, we propose, using the SSH protocol on a Linux system, a comparative study of our smart defensive honeypot Asgard [ 1 ], leveraging the RL algorithm Q -learning to address some of the limitations of these systems, with medium- and high-interaction honeypots.' The introduction contains some discussions on existing honeypots/state of the art, but it seems quite arbitrary, and it not really clear how or why this is relevant for the topic and the remainder of the paper. I don’t see that the classifications in Table 1 are referred to in the paper, and so it should just be removed. Instead, the authors relate to LiHP, HiHP, MiHP (in that order). The end of the introduction contains some overlapping objectives and contributions, which is a bit confusing. I suggest remove the redundancy.

An overall issue is that goals, purposes, and assumptions are often implicit or understated. 1) The paper lacks a clear problem statement. What is the actual goal of this paper? Just to make a comparison of four honeypots? 2) What is the premise and goal of a well-performing honeypot? Section 5.1 gives the impression that it is to receive the greatest number of commands. How is this meaningful, and how does this correspond with the informally stated goal of "revealing malicious techniques", line 31? Moreover, the statement 'These systems can intelligently adapt their behavior to match or to counter the attacker’s intention, which results in delayed interactions' is confusing. Is it a goal to match and to counter the attacker’s intention, or to 'delay interactions', what ever it means? How could a system 'know' the attacker's intention to begin with? Please explain what this means and why this is of significance. 3) It is not clear what the basis and the actual assumptions for the machine learning are. Is the goal of the ML to achieve a strategy/behavior so in average to receive the greatest number of commands? Something else? 4) Section 2 touches on theoretical aspects of ML, where terms such as 'reward' is just some value, but to the non-expert reader it is not clear in what sense this translates into practice as actual rewards to the attacker. 5) The distinction between 'attack episodes' and 'attacks' is unclear. Obviously, the number of attack episodes are much greater than the number of attacks.

I think the paper is vague when it comes to how adaptive honeypots perform. This should be better elaborated in sections 5.7, 5.8, and the conclusion.

Other:

- A related work-section is missing. What other comparative studies are there, and how are the findings in comparison to other work?

- Language: Sometimes incorrect the use of past tense, e.g., 'healthcare increased' -> 'healthcare have increased', line 21.

- The function E in Eq. 2 is not defined.

Author Response

Dear Reviewer 1,

First of all, thank you for your valuable comments and suggestions.

We understood that there were some confusions in the wording, and that we will reread the paper as a whole to clarify the contentious points.

In the meantime, you can find our response to your comments and suggestions in the attachment.

Reviewer 2 Report

  1. The research investigates four honeypots that represent three different classes of the honeypots and the experimental results show that Asgard can achieve a fairly good result in terms of the average number of commands compared to the conventional honeypots.

Suggestions:
1. line 135: Between "2. Background" and "2.1. Reinforcement learning", it should have some statements like: "In this section, we will describe....."
2. line 239: The title of section 3, "3. Methodology, Approach, and Proposed Models" is not suitable. Usually, "proposed models" mean the paper proposed model, not the other proposed models.
3. In the abstract, the paper says "the experimental results show that Asgard can achieve a fairly good result in terms of the average number of commands compared to the conventional honeypots," but in section 5.1 "5.1. Collected Attack Data" the paper says "Even though Cowrie attracted the most attacks, if we look at the 452 average number of commands that each honeypot received, Cowrie performed poorly, it 453 received only 5.51 commands on average compared to 18.48 commands for Aster, 9.68 for 454 Asgard and 8.2 for Midgard. Since most of the attacks were from automated programs and 455 were very targeted, they were very similar." So the conclusion is which one? "Asgard can achieve a fairly good result" or "they were very similar" needs to make sure.

Author Response

Dear Reviewer 2,

First of all, thank you for your valuable comments.

You can find our response to your comments in the attachment file.
